# Brief communication: Towards disability inclusive risk management

Vesna Coutureau[1], Tabea Bork-Hüffer[2], Margreth Keiler[1]

[1]Department of Geography, University of Innsbruck, Innsbruck, 6020, Austria
[2]Department of Geography, Heidelberg University, Berliner Str. 48, Heidelberg, 69120, Germany

*Correspondence to*: Tabea Bork-Hüffer (tabea.bork-hueffer@uni-heidelberg.de)

**Abstract.** People with disabilities face heightened vulnerability during disasters, yet they remain underrepresented in risk management planning and response. This brief communication reports findings from a pilot study in Tyrol, Austria, which assessed flood exposure and disaster preparedness in facilities serving people with disabilities. Spatial analysis revealed significant exposure to flood hazards, while qualitative interviews and document analysis identified critical gaps in risk awareness, preparedness, and inclusive planning. The study underscores the urgent need for disability-inclusive disaster risk management, emphasizing accessible information, targeted preparedness measures, and the integration of disability perspectives into emergency planning to enhance resilience for all.

## 1 Introduction

According to the WHO (2023), approximately 1.3 billion people globally—one in six individuals—experience a significant disability, making them the largest minority worldwide (UN DESA, 2024). According to the United Nations Convention on the Rights of Persons with Disabilities (UN CRPD, 2006), persons with disabilities includes those with long-term physical, mental, intellectual, or sensory impairments, as well as chronic illnesses, which—when interacting with various barriers—may hinder full and effective participation in society on an equal basis with others. The global commitment to "leave no one behind" was articulated in the 2030 Agenda for Sustainable Development, the UN CRPD (2006), and the EU Disability Strategy (EC, 2021), which further emphasizes socially just, inclusive, and sustainable transitions in Europe. However, progress in implementing these key conventions and agreements remains slow, and significant inequalities persist for people with disabilities worldwide.

People with disabilities are at particular risk during hazard events and emergency situations. For example, several studies across different contexts underline that people with disabilities are significantly overrepresented among fatalities related to flood events (e.g., Petrucci, 2022). The Sendai Framework for Disaster Risk Reduction (SFDRR) 2015–2030 (UNISDR, 2015) emphasizes the necessity of inclusive disaster risk management, explicitly advocating for the engagement of persons with disabilities in all disaster risk reduction (DRR) processes and especially in Priority 4 "Enhancing disaster preparedness for effective response, and to 'Build Back Better' in recovery, rehabilitation and reconstruction". It underscores the importance of accessibility and the elimination of discrimination to ensure effective participation of all societal groups. Stough and Kang (2015) evaluated the SFDRR's implications for persons with disabilities, underlining the importance of accessibility, inclusion,

and universal design in disaster risk reduction strategies. They also highlighted that the disability-related concepts in the SFDRR should serve as key guiding principles in the field.

However, as we argue in this commentary, people with disabilities remain persistently overlooked in disaster risk management, necessitating a fundamental shift toward inclusive risk management practices—especially in light of climate-change-related increases in hazards and risks (IPCC, 2022). Bennett (2020) assessed the implementation of the SFDRR's priorities with regard to people with disabilities, highlighting the need for enhanced disaster preparedness and a greater consideration of intersectional vulnerabilities. Villeneuve et al. (2021) propose a person-centered capability framework for Disability Inclusive Disaster Risk Reduction (DIDRR), emphasizing the importance of recognizing and enhancing the specific capabilities of individuals with disabilities in disaster contexts. The framework advocates for targeted, context-sensitive actions that address systemic barriers, empower individuals, and foster inclusive practices in the DRR process, particularly during emergency situations.

To substantiate our argument, we use a pilot study from the alpine state of Tyrol, Austria. It analyzed the proneness of facilities for people with disabilities to flood events and examined levels of awareness and preparedness—both within disability facilities, organizations, and support networks, and among representatives of disaster risk management and emergency responders.

## 2 Method and results

### 2.1 Case study Tyrol in Austria

The United Nations criticized Austria for lacking an inclusive, human rights-based disaster strategy (Amt der Tiroler Landesregierung, 2024). The National Action Plan on Disability (NAP) 2022-2030 aims to implement the UN CRPD in Austria, with a focus on strengthening federal cooperation and enhancing the participation of persons with disabilities (BMSGPK, 2022). Section 1.7 "Persons with Disabilities and Crisis Situations," specifically addresses inclusive disaster protection.

A key challenge remains the lack of data on affected individuals and their specific requirements, which hinders the provision of targeted assistance. To address this issue, efforts are underway to establish a more comprehensive data foundation. Simultaneously, an accessible emergency call and warning system will be implemented to ensure equal access to critical information. Additionally, persons with disabilities and their organizations will be integrated into crisis response teams and disaster planning processes to ensure that their requirements are adequately considered (BMSGPK, 2022). Furthermore, the NAP promotes knowledge exchange between civil protection authorities and disability organizations to refine existing measures. Technical innovations, such as the introduction of a crisis situation register, are also intended to optimize crisis management. The overarching goal of these initiatives is to enhance the resilience and safety of persons with disabilities in emergency situations (BMSGPK, 2022).

The Tyrolean Action Plan (TAP) seeks to implement the UN CRPD at the regional level. It emphasizes that people with disabilities face particular challenges in disaster situations and that existing crisis plans do not take them sufficiently into account (Amt der Tiroler Landesregierung, 2024). The aim is to ensure equal access to emergency care through barrier-free warning systems, appropriate equipment, and targeted training for emergency services. To this end, civil protection and disability organizations should work closely together, plans should be regularly updated and persons with disabilities should be more strongly involved in prevention to ensure that they are not excluded. Implementation will be carried out gradually until 2030 and includes specific measures in the areas of planning, alerting, evacuation, and accommodation. These measures defined in the Tyrolean Action Plan are based on a participatory process involving the civilian population and are structured according to short-, medium-, and long-term timeframes. In the long-term, the objective is to achieve genuine, equal participation (Amt der Tiroler Landesregierung, 2024). Disaster risk management in Tyrol is based on the Tyrolean Crisis and Disaster Management Act (TKKMG), which addresses all coordinated measures in the areas of prevention, preparedness, response, and recovery after events. The TKKMG categorizes disasters as local, cross-municipal, or cross-district damaging events, thereby delineating the responsibilities of municipalities, district administrations, and the provincial government. Mayors are responsible for drawing up municipal disaster management plans, while district administrations and the provincial government are required to draft corresponding plans for large-scale events. These plans include a comprehensive analysis of local conditions, potential hazards, warning and alert systems, evacuation measures, and available resources (TKKMG, 2024). The specific structure of these plans is regulated in the Disaster Protection Plan Ordinance. However, as this ordinance is not publicly accessible, no reliable information is available regarding its actual content.

Tyrol's disaster protection plans do not include specific provisions for people with disabilities, despite legal requirements (TKKMG, 2024). The Tyrolean Action Plan acknowledges this shortcoming and calls for legal adjustments as well as the regular inclusion of people with disabilities (Amt der Tiroler Landesregierung, 2024). Yet, reliable data on the inclusion of people with disabilities in disaster risk management in Tyrol remains insufficient. The 2022 micro census supplementary survey provides only limited insights, as it does not account for residents of facilities for people with disabilities (Schuller et al., 2024). River floods and torrent floods represent two primary types of flood hazards in mountain regions, each requiring distinct hazard mapping approaches due to their specific characteristics.

## 2.2 Methods

Given the limited research on the inclusion of people with disabilities in Austrian disaster risk management, this pilot study adopted an exploratory design that combined spatial mapping of work and residential facilities to identify flood-proneness with expert interviews and document analysis.

The first step involved the identification of work and residential facilities for people with disabilities in the state of Tyrol through the analysis of publicly available information on institutions and their locations. People with disabilities living in private households could not be included in the spatial analysis due to data protection requirements and the absence of a

registry of persons living independently. We note that while this provides important insights into populations living and/or working in institutions—which remain relatively prevalent in Austria and encompass people across all disability groups as outlined by the UN CRPD—it underrepresents those living and/or working independently. The main limitation of excluding people living in private households is that we do not know how many of them reside in potential flood-prone areas. However, the findings from the document analysis and qualitative interviews are also likely applicable to people living in private households.

A spatial analysis was conducted to assess the exposure of residential and workplace facilities to flood-prone areas of Tyrol, using hazard maps for river and torrential flooding, as well as modelled flood hazard zones (BML, 2025). In Austria, three primary sources of flood hazard information are available. Hazard mapping for torrent floods, including debris flows, is conducted at the local scale (1:2000 to 1:10,000) by the Austrian Service for Torrent and Avalanche Control (WLV). These maps delineate areas affected by design events with a return period of 150 years. The Federal Water Engineering Administration (BWV) provides the hazard maps for river flooding, primarily considering events with a return period of 100 years. The third source is based on flood modeling, which estimates 1-in-30 (HQ30), 1-in-100 (HQ100), and 1-in-300 (HQ 300) year flood events, in line with requirements of the EU Floods Directive 2007/60/EC (EC, 2007).

To complement the spatial analysis, the study employed a qualitative exploratory approach based on semi-structured expert interviews. Experts were defined as individuals with specialized role knowledge relevant to disaster risk management or disability services. Interviews were conducted with two representatives from disaster management, four employees of disability service facilities, and one researcher specialising in disability-inclusive disaster risk reduction. While the number of interviews was limited, the aim was to capture a diversity of perspectives from key stakeholder groups. The findings should therefore be seen as exploratory in nature and not intended to be representative of all actors in the field. All interviews were transcribed and coded using MAXQDA software. The data were analyzed through qualitative content analysis following Kuckartz (2018), which allows for a systematic and transparent process of categorization, combining deductive codes derived from existing frameworks with inductive codes emerging from the data. In addition to the interviews, four protocols documenting the implementation processes of the Tyrolean Action Plan were analyzed. These were examined using the same deductive categories applied in the interview analysis (Amt der Tiroler Landesregierung, 2024).

## 2.3 Results

In the study area we identified 84 workplace facilities and 74 residential facilities of people with disabilities.

### 2.3.1 Exposure to floods

A total of 31 work facilities and residential facilities for people with disabilities are located, at least partially within flood-prone areas. A differentiated analysis reveals that three work facilities are situated in areas of high hazard (HQ30), while twelve work facilities and twelve residential facilities are located in moderate-hazard areas (HQ100). Figure 1 presents selected

sections of Hall in Tirol and Schwaz, highlighting residential and work facilities at risk within the flood zone. Approximately 102 people with disabilities live in the exposed residential facilities and 140 people with disabilities are employed in those workplaces. The majority of facilities fall within low-hazard zones (HQ300), including 15 work facilities and eleven residential facilities.

### 2.3.2 Flood risk awareness and preparedness

Awareness and sensitization of the population are crucial for an appropriate and effective response to disasters. Experts note that there is a lack of societal awareness regarding flood risk (expert in disaster risk management). In facilities for people with disabilities, there is often a lack of awareness of exposure to natural hazards such as flooding (experts in disability matters), even though publicly accessible flood maps provide this information. While fire protection measures are well-established, interviews revealed that employees in disability services facilities were not aware of the flood risks to their facilities: "I think even though we're in such a zone, I actually feel totally safe here" (expert in disability matters). Although Lebenshilfe Tyrol has a risk management team, the hazard and exposure situation of individual locations is only partially known and is not centrally documented (expert in disability matters). Systematically recording this data would be essential for effective flood protection and preparedness.

A significant challenge for people with disabilities is the limited accessibility of information on natural hazard risks, as such information is often not presented in accessible formats. For example, while the Water Information System Austria (WISA) map offers crucial data on flood risks, its website is not fully accessible (BML, 2025). As the document analysis showed, the Austrian Civil Protection Association's information is not available in accessible formats and fails to account for the requirements of people with disabilities, thereby hindering their participation in disaster preparedness measures. According to the Global Survey Report on Persons with Disabilities and Disasters (UNDRR, 2023), many individuals criticize the lack of accessible information, which reduces risk awareness and complicates participation in protective measures. As a result, the perspectives of people with disabilities are often inadequately addressed in disaster management planning.

Another important issue in this context is the specific requirements for inclusive early warning systems. Historically, these systems have relied on acoustic signals, such as sirens, which are not accessible to all. There is an urgent need to design early warning systems that effectively reach all individuals and clearly communicate necessary actions. Currently, the AT Alert mobile phone warning system is being tested as a more accessible alternative (Amt der Tiroler Landesregierung, 2024). Additional studies reveal that reliable data on people with disabilities in disaster risk management in Tyrol remain insufficient. The 2022 micro census supplementary survey provides only limited insights, as residents of facilities for people with disabilities are not included (Schuller et al., 2024). To overcome this lack of data, a voluntary emergency register enables self-registration but is viewed critically. Participants emphasized concerns about the collection of sensitive data, such as information about disability type and medical requirements, as well as limited trust in whether such registers would be effectively used in practice (expert in disaster risk management Tyrol). Nevertheless, such a register could facilitate more

targeted resource allocation during disasters. It "would be a huge advantage if the emergency personnel knew where people with disabilities live, what their needs are, and what assistive technologies they use" (member of the Austrian Disability Council). In general, our pilot study highlights once again that people with disabilities remain underrepresented in emergency response organizations (see also Gabel and Schobert, 2024). Legal barriers, such as health suitability requirements, contribute to this exclusion. People with disabilities can actively contribute to disaster response, provided that accessible frameworks are in place (expert in disability inclusive disaster risk reduction). However, preparedness materials and training programs are often not fully accessible. Neighborhood assistance plays a key role but requires coordinated state support (expert in disaster risk management Tyrol).

Improving knowledge about disaster preparedness is essential for strengthening long-term self-help capacities. Inclusive, data-driven disaster risk management remains a key challenge for Tyrol. The integration of people with disabilities, accessible information dissemination, and the establishment of robust legal frameworks must be consistently advanced to ensure safety and participation for all.

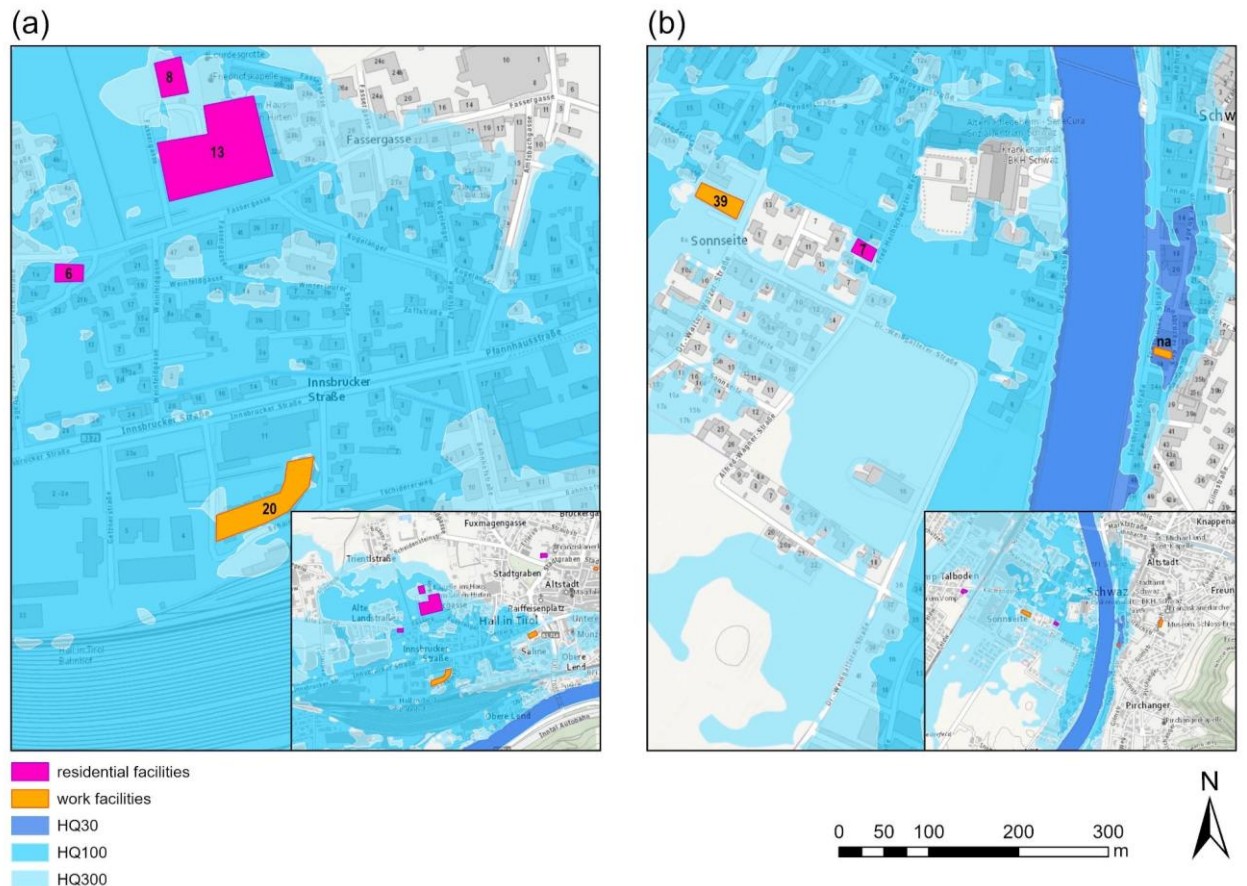

**Figure 1:** Residential and work facilities for people with disabilities exposed to flood risk (HQ30 – high, HQ100 – moderate, and HQ300 – low) in (a) Hall in Tirol and (b) Schwaz. Numbers indicate the number of people with disabilities in the facilities. Data sources: data.gv.at and basemap.at

## 3 Discussion and Conclusions

As the UN DESA (2019) has noted: "[s]ustainable development for all can only be attained if persons with disabilities are equally included as both agents and beneficiaries as countries strive for a sustainable future". When it comes to risk management, as the case study in this article has shown, people with disabilities remain underrepresented. The study underscores significant gaps in inclusive disaster risk management in Tyrol, particularly in risk awareness, preparedness, and data accessibility which are needed to be addressed for inclusive disaster risk management and aims to enhance the resilience and safety of people with disabilities.

Based on the analysis of identified barriers, five key recommendations emerge: (1) Raising public awareness about flood risk and the specific requirements of people with disabilities is essential. This can be achieved through workshops, accessible public awareness campaigns, and joint exercises involving emergency response organizations and the community. (2) Enhancing self-help capacities of people with disabilities is critical. This includes the provision of accessible informational materials, inclusive first aid training, and the removal of barriers in early warning systems and evacuation processes (Gabel and Schobert, 2024). (3) Improving data collection to enable targeted planning and assistance, such as through emergency registries that include facilities serving people with disabilities but also people living independently, while ensuring the protection of sensitive data is central. These systems should be co-designed with disability organizations. (4) Ensuring accessible disaster risk communication, e.g., preparedness information, which should be available in easy language and across various media formats. (5) Promoting the inclusion of people with disabilities in emergency response organizations as a key measure to leverage their resources, promote diversity, and reduce biases (Gabel and Schobert, 2024). Implementing these measures can foster individual safety but also contribute to a more inclusive society as a whole.

Importantly, although legal frameworks and governance structures are regionally specific, many of the issues identified—such as inaccessible early warning systems, lack of facility-level awareness, and limited participation of people with disabilities—are highly relevant for other mountainous regions in Europe (e.g., in Switzerland, Northern Italy, or Southern Germany). These areas face similar combinations of topographic hazards (river and torrential flooding), dispersed settlement patterns, and fragmented governance. Thus, our findings can inform broader debates on disability-inclusive disaster risk management across Europe.

Implementing effective inclusive risk management remains complicated and challenging due to the diverse nature of disabilities, each requiring tailored measures and responses. However, adopting an inclusive approach not only addresses the specific requirements of people with disabilities but also strengthens the resilience of other vulnerable groups. These groups include older adults, individuals with temporary illnesses or injuries, those with mobility impairments, pregnant women, and children and young people along with their families. By fostering a comprehensive and inclusive risk management strategy, communities can enhance overall preparedness and ensure that no one is disproportionately affected during damaging natural hazards events.

Beyond this pilot study that focused on flood-proneness, institutional perspectives, and structural gaps, more comprehensive analyses are needed, ideally adopting a participatory or co-productive approach that integrates people with diverse disabilities including those that live in facilities but also independently throughout all stages of the research.

*Data availability*: The flood risk areas are available via data.gv.at and are publicly accessible. Data on facilities for persons with disabilities are not publicly accessible due to data protection regulations.

*Author contribution*: Conceptualization, A.C., T.B.-H., and M.K.; methodology, A.C., T.B.-H., and M.K.; validation, A.C., T.B.-H., and
210 M.K.; formal analysis, A.C.; investigation, A.C.; data curation, A.C.; writing—original draft preparation, A.C., T.B.-H., and M.K.; *Review and editing*: T. B.-H. and A.C.; Supervision: T.B.-H. and M.K

*Acknowledgments*: The authors thank the participants involved in this project.

*Competing interests*: Margreth Keiler is member of the editorial board of Natural Hazards and Earth System Sciences. All authors have read and agreed to the published version of the manuscript.

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
