# Peer review of "Brief communication: Towards disability inclusive risk management"

_EGUsphere, 2025_

## Author Response (AR1)

Dear anonymous reviewers,

Many thanks for the very supportive feedback and thorough engagement with our article. We feel greatly encouraged in both our conceptual approach and the empirical study. Please find below our more detailed responses to your helpful comments. We have copy-pasted each comment into this form and respond in detail right below it.

COMMENT 1 (reviewer 1)

The paper is generally well structured and well written and presents an important topic within the field and with revisions would make a useful contribution to the literature on disability inclusive disaster risk reduction in relation to early warning systems. This would require further justifications for the choices made and discussion on the limitations of these choices, at it stands there are weak or missing justifications for critical methodological decisions. The recommendations that are given in the results sections have been made on multiple occasions in other work, in order to strengthen the papers unique contribution it would be important to see this further linked to the empirical evidence that was collected, this would help to provide a more novel perspective - from a case study in this region. In addition, the results section is mostly made of recommendations rather than empirical evidence, which leaves the reader wanting more. Tying these up in a conclusion section would benefit the paper and strengthen the overall argument of the piece, by not having a conclusion section the conclusions are not substantial. It is also important to be clearer with terms, for example 'residential facilities', equally in the final sentences, resilience is introduced but without clarity.

RESPONSE 1:

We thank the reviewer for these valuable suggestions. We have expanded the Methods section to better justify our methodological decisions, particularly the focus on residential and workplace facilities. We explain that these facilities were chosen due to data availability. We have acknowledged limitations of this choice such as the exclusion of private households.

We have more clearly differentiated the results section from the discussion and conclusions section.

COMMENT 2 (reviewer 1)

In the introduction it would be useful to provide further comment on some of the work that has taken place on disability inclusive DRR to orientate the reader as there is an increasing body of work on this topic. This includes emerging research on early warning systems for people with disabilities in the context of flooding.

RESPONSE 2:

Thank you for this comment. We agree; however, the journal imposes a limit on the number of references allowed in commentaries. For this reason, we had to substantially reduce the references prior to the first submission and were only able to add a few others during the revisions by replacing some of the existing ones. Therefore, we use this space in the responses to provide additional references: In

addition to the frameworks already mentioned, frameworks such as the Dhaka Declaration on Disability and Disaster Risk Management (Dhaka Conference on Disability & Disaster Risk Management, 2015) have provided guiding principles for inclusive DRR practices. The Gaibandha Model offers a practical example of community-based inclusive DRR. The UNDRR provides a checklist and implementation guide for inclusive early warning and early action (UNDRR, 2023). Complementing existing research, the dissertation of Gabel (2025) examines the disproportionate risks faced by persons with disabilities in disasters and proposes a conceptual framework that integrates disaster research, disability studies, and applied ethics to inform inclusive and reasonable accommodation practices in disaster risk management. Chisty et al. (2021) evaluate the disability inclusiveness of flood early warning systems in Bangladesh, finding that while people with disabilities have risk awareness, gaps in monitoring, communication, and especially response capabilities limit their ability to act on warnings.

COMMENT 3 (reviewer 1)

Did the research carry out participatory methods or is the process of the local government participatory? In the case study section it mentions participatory methods but it is not clear if this is in relation to this study or not.

RESPONSE 3:

This study did not use participatory methods. However, participation is taken into account during the creation of the Tyrolean action plan and its implementation processes. We clarified this in the paper.

COMMENT 4 (reviewer 1)

How did the paper reach a saturation point with a small number of qualitative interviews? Were these long in-depth interviews and perhaps more than one with each individual, what where ther content and aim? How were these experts selected? Who was the "expert in the field" - in what field DRR or Disability or disability inclusive DRR? What was the justification for not including people with disabilities themselves? The decision-making processes behind this research design need to be more clearly outlined and with limitations provided.

RESPONSE 4:

We would like to note that this is a commentary about a pilot study. We have revised the methods section to provide more transparency about research design. The qualitative semi-structured interviews were in-depth, focusing on awareness of flood risks, preparedness measures, challenges to inclusion, and cooperation between disability facilities and disaster management actors. The experts were purposefully selected based on their roles. We reached some thematic saturation despite the small number of interviews, as key themes (e.g., lack of accessible information, insufficient legal frameworks, and missing inclusion in planning) were consistently repeated across participants. Given that this was a pilot study, our aim was to explore institutional perspectives and structural gaps.

Still, as we added to the conclusions, further and comprehensive research beyond this pilot study is needed. Pivotally, as we now also note at the end of the commentary, this would need to include people with disabilities and ideally pursue a participatory or co-productive design.

COMMENT 5 (reviewer 1)

At points the paper is repetitive and would benefit from editing and diversity in its vocabulary. For example, it says in multiple locations that training programs, early warning systems etc are not accessible and 'overlook people with disabilties' - 35, 160, 135, 65.  In what ways and for what individuals? This leaves some of these observations to feel a little vague - further detail would be beneficial, and for this to be reframed or re-worded differently each time.

RESPONSE 5:

Disabilities are extremely diverse, which is why the possibilities for overlooking people with different disabilities in disaster management are also very high. This particularly affects the type of communication, technological gaps, physical barriers, and, of course, overlooking people with disabilities' needs in planning.

Thank you for pointing out the word repetitions. We have revised and diversified this.

COMMENT 6 (reviewer 1)

The paper would benefit from further discussion on the challenges of defining disability and the multiple ways it can be understood and defined, this is mentioned in the final results but not in the outset of the paper. This would be beneficial in the sense that it would allow the paper to reflect on and provide clarity on how residential and workspace places of people with disabilities were identified. Did residential facilities include private residencies i.e private homes? Further clarity of this definition would be helpful as well as why these were selected. From line 125 it suggests this means residential facilities i.e homes for people with disabilities, this needs clarity earlier in the paper but also a justification for why these are the focus rather than general households. How does this influence the findings or importance of the study as this was critical to its design.

If also residential houses i.e private home, How was disability defined within this identification process and did this account for less severe disabilities, or neurodiversity for example with ADHD or autism? As all of these would come under the explanation of disability in the CRPD provided in the introduction. - what disabilties or individuals with disabilties is this study relevant to and what are the limitations if focussing on residential homes for people with disabilties (if meaning those who cannot live alone) and what was is justification and/or importance of this choice and this group?

RESPONSE 6

Defining disability in multiple ways:

Due to data protection and no available registry for people with disabilities living in private households could not be included in the study. We acknowledge that this excludes a substantial

proportion of people with disabilities from the spatial analysis. At the same time, all residential facilities in Tyrol were included in the spatial analysis, as their exact building locations could be obtained and intersected with flood hazard zones. Residents and employees in these facilities represent a broad variety of people with disabilities, encompassing all groups outlined in the UN CRPD. Therefore, the pilot study does not systematically exclude groups with specific disabilities. The main limitation of excluding people living in private households is that we do not know how many of them reside in potential flood-prone areas. However, the findings from the document analysis and qualitative interviews are also likely applicable to people living in private households. Disability was defined, as stated in the introduction, to include all relevant groups, ensuring that all categories of disabilities were considered in the study.

COMMENT 7 (reviewer 1)

145 line explains that the early warning systems was viewed critically by a participant, in what way, was this probed, can further information be provided? Being more specific would provide greater clarity and orientation to the reader and strengthen a core argument of the piece, this may also help to provide more specific recommendations, including context specific recommendations that could then be broadened out to wider arguments or suggestions.

RESPONSE 7

We have expanded the relevant section to specify that the critical views focused on concerns regarding data protection, privacy and limited trust in whether such registers would be effectively used in practice. We now explain that participants noted potential reluctance of persons with disabilities to disclose sensitive information and highlighted uncertainties around data handling.

COMMENT 8 (reviewer 1)

In line 180 it says 'inherently complicated and challenging', why is it 'inherently' complicated rather than complicated because of institutional and/or access, cultural barriers etc, is there any empirical evidence from this study to support this statement?

RESPONSE 8

Thank you. Indeed, there was no basis to describe this as inherently complicated, so we have removed that wording. The empirical findings clearly underline the multidimensionality and complexity of disaster risk management. We have further expanded this section and believe it now communicates these points more effectively.

COMMENT 9 (reviewer 1)

Technical errors:

The CRPD provides and explanation of disability rather than a definition because of the reality of the multiple ways of which it can be defined. In line 180 it says inherently complicate and challenging, this is a typo.

The figures are too small, they cannot be read easily which means they were not supporting the papers arguments well and present an inclusion challenge in its own right.

The Global disability report should be referenced as UNDRR rather than Gvetadze et al., 2023. This should be cited as the Global disability report on disasters not the global disability report which is written by the WHO and is a different report. It might also be useful to emphasise that the difference between the 2013 and 2023 report statistics are limited to further accentuate the point and need for disability-inclusive early warning systems.

While the topic is important and the paper has potential, it needs further clarification, restructuring and methodological transparency.

RESPONSE 9

This is absolutely right, the UN CRPD does not define disabilities but enlists groups included in its understanding of disabilities. We have changed this and also addressed the other technical issues. The Global Survey Report is now correctly cited as UNDRR (2023).

COMMENTARY 10 (reviewer 2)

This paper studies an important issue by examining the inclusion of people with disabilities in disaster management, offering relevant findings and practical recommendations. However, revisions are needed, specifically in describing methods and results.

Title, Abstract, Introduction

The title could be more specific by indicating that the paper focuses on the inclusion of people with disabilities, rather than inclusion in general. The abstract provides a clear and concise overview of the paper. However, it should also mention the third method used—document analysis—to ensure completeness. The literature cited is primarily policy-based, with little reference to scientific research. You might want to review more scientific sources or justify why it is not necessary in this case.

RESPONSE 10

Thank you for highlighting the need of a more specific title, we clarified that the paper focuses on disability inclusive risk management and added the third method that has been used throughout the study.

We mainly used policy-based literature in our pilot study as we had a strong regional focus on Tyrol. Our goal was to identify gaps, but also to examine existing measures for disability inclusive disaster risk management. There is hardly scientific literature available with this regional focus, which is also why we want to contribute by publishing the findings of this pilot study.

COMMENTARY 11 (reviewer 2)

Methods

The description of the methods could be clearer, particularly regarding why and how you chose your data sources and how the analysis was conducted.

Data sources: The interviewee profiles could be clarified. It is not clear whether "a researcher in this field" refers to disability research or another discipline. The methods section refers to "employees in disability service facilities," whereas the results mention "facility managers." Also, the sample size is relatively small, so this limitation should be acknowledged or justified. The process for selecting documents for analysis should also be explained, along with a brief description of the content of these documents. The interview guide and the analyzed documents could be attached to an appendix to increase transparency.

Analysis: More detail on the coding process for interviews would be useful, including presenting the deductive codes you used and the inductive codes that emerged during analysis. Furthermore, it is not clear how you analyzed the documents about the implementation process of the Tyrolean Action Plan. Did you use the codes from the interview analysis, or did you develop new codes? Also, it would be useful to explicitly state which data and methods were used to answer which research questions.

RESPONSE 11

Data sources: We clarified that the researcher is specialized in disability inclusive disaster risk reduction. We have also standardized the terminology and now only refer to "employees in disability service facilities".

While the number of interviews is limited in this pilot study, the focus was on capturing a range of perspectives from key stakeholder groups. The findings should therefore be seen as exploratory in nature and not intended to be representative of all actors in the field.

Analysis:

The documents about the implementation process of the Tyrolean Action Plan were analyzed with the same codes as the interviews. In an additional document we attached the codes that we have used for our analysis.

COMMENTARY 12 (reviewer 2)

Results

The caption for Figure 1 could include another half-sentence explaining the flood intervals (HQ30, …). In the results section, the focus should be on the authors' own findings. References to theory or other studies should be moved to Introduction or Discussion. This section could be more comprehensive and include more findings and richer detail; for example, you could include quotes from the interviews or the analyzed documents that exemplify key results.

RESPONSE 12

The caption of Figure 1 now explains HQ30, HQ100, and HQ300 intervals (high, moderate, and low flood risk). We have included two quotations to exemplify our key results.

COMMENTARY 13 (reviewer 2)

Discussion and Conclusions

The final section should be titled "Discussion and Conclusions" rather than "Results" (as there is another Results section before). The conclusions are strong and actionable, offering five specific recommendations to improve the inclusion of people with disabilities in disaster management planning and preparedness. It might be interesting to reflect on what role the choice of case study played; do you expect similar results in other circumstances or was this case study one of a kind? Are there any best practice examples that you could cite? It might also be interesting to reflect on or search your data for mentioned barriers that prevented better inclusion of people with disabilities in your case studies. Overall, the paper is well-written and offers valuable insights into the inclusion of people with disabilities in disaster management; with more methodological clarity, richer presentation of results, and minor structural changes it could make a valuable contribution to the field.

RESPONSE 13

The final section is now titled "Discussion and Conclusions" and we made the transferability of our findings clear. The results are not just relevant to Austria, but can inform DIDRR in other mountainous regions in Europe and even beyond, despite differences in legislation.